# Climatic Alterations Influence Bacterial Growth, Biofilm Production and Antimicrobial Resistance Profiles in *Aeromonas* spp.

**DOI:** 10.3390/antibiotics10081008

**Published:** 2021-08-20

**Authors:** Miguel L. Grilo, Ana Pereira, Carla Sousa-Santos, Joana I. Robalo, Manuela Oliveira

**Affiliations:** 1Centro de Investigação Interdisciplinar em Sanidade Animal (CIISA), Faculdade de Medicina Veterinária, Universidade de Lisboa, 1300-477 Lisbon, Portugal; 2Marine and Environmental Sciences Centre (MARE), Instituto Universitário de Ciências Psicológicas, Sociais e da Vida (ISPA), 1100-304 Lisbon, Portugal; ana_pereira@ispa.pt (A.P.); carla.santos@ispa.pt (C.S.-S.); jrobalo@ispa.pt (J.I.R.)

**Keywords:** microcosm, *Aeromonas*, climate change, temperature, pH, biofilm, antimicrobial resistance, water

## Abstract

Climate change is expected to create environmental disruptions that will impact a wide array of biota. Projections for freshwater ecosystems include severe alterations with gradients across geographical areas. Life traits in bacteria are modulated by environmental parameters, but there is still uncertainty regarding bacterial responses to changes caused by climatic alterations. In this study, we used a river water microcosm model to evaluate how *Aeromonas* spp., an important pathogenic and zoonotic genus ubiquitary in aquatic ecosystems, responds to environmental variations of temperature and pH as expected by future projections. Namely, we evaluated bacterial growth, biofilm production and antimicrobial resistance profiles of *Aeromonas* species in pure and mixed cultures. Biofilm production was significantly influenced by temperature and culture, while temperature and pH affected bacterial growth. Reversion of antimicrobial susceptibility status occurred in the majority of strains and tested antimicrobial compounds, with several combinations of temperature and pH contributing to this effect. Current results highlight the consequences that bacterial genus such as *Aeromonas* will experience with climatic alterations, specifically how their proliferation and virulence and phenotypic resistance expression will be modulated. Such information is fundamental to predict and prevent future outbreaks and deleterious effects that these bacterial species might have in human and animal populations.

## 1. Introduction

Environmental conditions are a major driver of bacterial activity and can shape the expression of several metabolic pathways [1,2]. Namely, such parameters have the potential to influence bacterial virulence (e.g., biofilm formation) and antibiotic resistance signatures [3,4].

Climatic scenarios, as predicted by simulation methodologies based on different levels of emissions, are projected to significantly differ from currently observed meteorological conditions [5]. Regarding aquatic ecosystems, and particularly in freshwater habitats, various environmental parameters are expected to be altered in the coming years. Water temperature, directly influenced by air temperature, is expected to rise across different habitats [6]. Additionally, the occurrence of heatwaves will likely increase, resulting in extended periods of drought associated with a low flow of freshwater systems, a decrease in water level and in dissolved oxygen concentrations [7,8,9]. Consequently, reduced dilution of freshwater streams will also affect ion balance levels [10,11]. These biotic changes will impact ecosystem dynamics and promote disruptions in species equilibrium [9,12]. All of these events are expected to significantly decrease freshwater’s quality [13]. Ultimately, these changes compromise future water availability, freshwater ecosystems’ structure and populations’ sustainability [14,15,16].

Natural aquatic ecosystems, often the last destination of terrestrial runoffs, are known reservoirs of both antimicrobial resistance and bacterial virulence determinants [17]. The microbiota present there, with or without direct connection with clinical infections, constitute a pool of information to the terrestrial microbiota or can even be disseminated to anthropogenic cycles [18]. This intricate connection between environmental microbiota and bacterial genus with effects at the One Health level stresses the importance of close surveillance of antimicrobial resistance and virulence dynamics in natural habitats in order to prevent epidemic situations both in anthropogenic settings and natural habitats [19,20]. Since modeling bacterial responses to changing environmental parameters in natural habitats is challenging, lab simulations—e.g., microcosm assays—are an important tool to predict how microbiota will respond to environmental cues foreseen in climatic predictions [21,22].

We hypothesize that aquatic bacteria’s antimicrobial resistance signatures and virulence traits, as well as their growth, may vary with changing environmental conditions. In order to test this, we applied microcosm simulation assays using different water temperatures and pH values following established emissions scenarios [5] to *Aeromonas* spp.—a model bacterial genus ubiquitous across different aquatic ecosystems—and evaluated changes in the antimicrobial resistance profile, biofilm production and growth of the isolates under study.

## 2. Results

Biofilm production by each of the *Aeromonas* strains in pure and mixed culture in the different assays is illustrated in Figure 1. Each strain’s response to temperature and pH was variable between species and within the same species.

When considering results by groups (*Aeromonas* species individually and mixed cultures), biofilm production in the mixed cultures’ wells was significantly lower (*p* < 0.001) than in the other groups. Additionally, water temperature also significantly influenced biofilm production (*p* = 0.006), with isolates exposed to the Fluctuations treatment producing less biofilm (Figure 2). The different pH conditions tested did not influence biofilm production.

Regarding mixed culture wells, re-isolation and identification of the initial *Aeromonas* pool added to each well was not possible with several combinations of temperature and pH treatments. *Aeromonas* species prevalence at the end of microcosm assays varied across the applied treatments and also between replicates (Figure 3). When evaluating the influence of each individual *Aeromonas* species present in mixed cultures on the biofilm production, it was observed that no species had a significantly different influence.

Some differences were observed regarding the growth of the isolates during the experiment (Figure 4). Significant differences were recorded between the tested *Aeromonas* species (*p* < 0.001). *A. veronii* isolates presented significantly lower concentrations than the other single and mixed cultures, while *A. hydrophila* presented significantly lower concentrations than *A. media* and mixed cultures. Temperature (*p* < 0.001) and pH (*p* = 0.007) treatments also influenced bacterial growth. While bacterial growth did not differ between current and fluctuations treatments, it was significantly increased in the RCP 4.5 treatment and decreased in the RCP 8.5 treatment. Bacterial growth was increased in acidic pH conditions (6.31) when compared to alkaline pH (8.61). Specific associations were also found between *Aeromonas* species and pH (*p* = 0.002) and between temperature and pH (*p* < 0.001). While *A. media* and mixed cultures presented higher concentrations in water microcosms with pH 6.31, *A. caviae* presented higher concentrations at pH 8.61. No differences were observed at pH 7.61. Regarding the interaction between temperature and pH, concentrations in the RCP 4.5 treatment were higher at pH 6.31, decreasing until pH 8.61. For RCP 8.5, higher concentrations were observed at pH 8.61.

The bacterial concentration was not correlated with biofilm production (*r_s_* = 0.020, *p* = 0.676).

Several changes regarding the antimicrobial resistance profile were observed among treatments for the same isolate (Figure 5). Observations between the control treatment (T0, pH 7.61) were similar to results obtained with the current treatment and similar pH levels. Phenotype variation occurred in a strain-dependent way, and it was specific for each antimicrobial compound tested. For all strains and antibiotics (except *A. hydrophila* and tetracycline), modification of the original susceptibility category occurred with at least one combination of treatments.

In certain situations, reversion of non-wild-type to a wild-type phenotype occurred only with specific combinations of temperature and pH. This is the case of erythromycin susceptibility and *A. caviae*, *A. hydrophila* and *A. media*. Regarding *A. caviae* and *A. media*, the same treatment (i.e., Current and pH 6.31) caused this phenomenon. In other cases, several combinations resulted in this reversion with no obvious pattern. The opposite was also observed (conversion from wild-type to non-wild-type) among the isolates. Although some treatments seemed to result in this situation more often for some antimicrobial compounds (i.e., RCP 4.5), a high variability was observed.

## 3. Discussion

Investigating how bacteria will evolve with environmental cues using natural habitats is a difficult task. Instead, the use of microcosm simulations allows the exploration of such associations, ensuring experimental control and uniformity. This methodology represents a first step in the prediction of transformations to occur in important bacterial genus with an impact at the One Health level, such as *Aeromonas* spp., and prepare for future outbreaks or phenotypical changes with consequences to public health. In this study, we show that different *Aeromonas* species adapt their growth, biofilm production and antimicrobial resistance signatures to environmental projections related to climatic alterations (i.e., temperature and pH) in water, highlighting the role that future climatic events will have in shaping bacterial activity, as well as virulence and resistance expression. It is noteworthy that, in this study, differences regarding growth, biofilm production and antimicrobial resistance signatures were observed using relatively small temperature and pH amplitudes, which are more likely to reflect future climatic trends.

### 3.1. Biofilm Production

In general, the studied isolates presented variability in the production of biofilm when exposed to the different temperature and pH treatments. Although some response patterns were present, the disparity in results between isolates of different species and within the same species highlights the fact that individual characteristics will govern how an isolate will respond to environmental cues; however, significant associations were observed. Mixed cultures produced significantly less biofilm when compared to the *Aeromonas* species individually. At the end of the microcosm assay, it was not possible to isolate all *Aeromonas* species in many mixed cultures. Some species absence was more evident than others (e.g., *A. caviae* in mixed culture #3 along the various temperature and pH treatments), although a general pattern was not present. Additionally, and while pH treatments seem not to influence biofilm production significantly, temperature influenced biofilm production in *Aeromonas* spp. Namely, isolates exposed to temperature oscillations (i.e., Fluctuations) produced less biofilm. Such biofilm production was not dependent on bacterial concentration. Distinct *Aeromonas* species display specific preferences regarding environmental parameters [23,24]. Although *Aeromonas* spp. possess stress response mechanisms to deal with environmental oscillations [25], they still impact several aspects of bacterial life. If the combined temperature and pH conditions fall within the optimal range for multiplication and virulence expression for each isolate, they will dictate the isolate’s competitiveness and ability to survive in an environment composed of multiple species [26,27]. Further, the level of interspecific competition for the limited resources will also hinder each isolate’s ability to allocate nutrients to processes such as biofilm production, contrary to what occurs in pure cultures [28]. Finally, environmental oscillations of abiotic factors, such as temperature, will create additional disturbances for the bacterial communities [22] and the overall combination of external stressors with internal competition is likely to impact the final biofilm production.

### 3.2. Bacterial Growth

In this study, a disparity in bacterial growth during the microcosm experiments was observed between the studied species (both in pure culture and in mixed culture). Overall, *A. veronii* isolates displayed a significantly lower growth when compared with other tested groups. Delamare et al. [29] highlighted lower growth patterns by *A. veronii* when compared to other *Aeromonas* species (i.e., *A. hydrophila*, *A. media* and *A. caviae*). Growth rate variability is a consequence of phenotypic diversity in bacteria [30]. Such variability can be the result of the nutrient uptake rate by the bacterial cell and of the resource’s distribution between the processes occurring in the bacterial cell [31]. *Aeromonas* strains and species growth variability likely reflect different limitations in these processes among the isolates, which can also explain differences observed not only for *A. veronii* but also for *A. hydrophila*. In the mixed cultures group, this pattern was not observed, and two hypotheses can be drawn: either other *Aeromonas* species present in the culture compensated for lower growth rates by *A. veronii*, or interspecific competition eliminated *A. veronii* presence in the microcosm wells (as stated in Figure 3), facilitating the growth of other species or canceling the growth effect *A. veronii* had in the total growth.

Both changes in water temperature, as well as in the pH conditions, played a significant role in the growth of *Aeromonas* spp. Regarding temperature, a biphasic effect was observed: while small increments in water temperature (i.e., RCP 4.5) seem to benefit the *Aeromonas* species under study, both in pure and in mixed cultures and favor their proliferation; once reaching a certain threshold imposed by higher temperatures (i.e., RCP 8.5), such boosting effect is lost and bacterial growth is lowered. Temperature is a determinant in bacterial growth and *Aeromonas* typically increase both growth and metabolic activity and decrease lag phase when experiencing higher environmental temperatures [32,33]; however, such growth reaches a plateau with temperature increments and starts to decrease before reaching maximum thermal tolerance [34], highlighting the role of thermal stress as a regulator of bacterial growth. It is noteworthy that, although cultures subjected to the fluctuation treatment experienced similar temperature values, such as the ones in the RCP 8.5 treatment, alternate exposure to higher (24.5 °C) and lower (21 °C) temperature values likely created buffer periods in which bacterial cultures could stabilize and multiplicate.

Regarding pH, the overall growth of *Aeromonas* spp. was higher in acidic environments when compared to alkaline environments. While some authors found a non-significant effect or a negative effect of pH on *Aeromonas* growth [33,35,36], *Aeromonas* are evolutionarily adapted to low pH environments, such as the gastrointestinal environment, and have built cellular responses (i.e., protective protein synthesis) that allow for acid tolerance [37]. Additionally, when exposed to acidic environments, the lag phase in *Aeromonas* is significantly shorter, prompting the beginning of the following growth phases sooner [38]; however, it is likely that different *Aeromonas* species display specific niche preferences and have evolved towards tolerance in different pH gradients. This explains why in this study some groups exhibited higher growths in acidic treatments (*A. media* and mixed cultures), while others performed better in alkaline pH (*A. caviae*). Additionally, both temperature and pH seem to play an interactive role, conditioning higher growth of *Aeromonas* spp. with specific combinations (i.e., RCP 4.5 and acidic pH, RCP 8.5 and alkaline pH).

### 3.3. Antimicrobial Resistance Profiles

Climate change has been implicated as a factor involved in increasing levels of antimicrobial resistance among different bacterial species in prolonged temporal sets. Distinct spatial patterns occur globally and are connected with local climacteric variability, highlighting how distinct geographical areas will be impacted by this problem in different proportions [39,40]. Specifically, regions expected to be more vulnerable to climacteric alterations are also the ones predicted to accumulate the highest prevalence of antimicrobial resistance [41]. Some authors report the role of increasing temperatures over time in the overexpression of this phenomenon in species such as *Escherichia coli*, *Klebsiella pneumoniae*, *Pseudomonas aeruginosa* and *Staphylococcus aureus* [40,42]. In a meta-analysis with isolates collected in aquacultures conducted by Reverter et al. [41], a similar conclusion was drawn for bacterial genera commonly infecting aquatic animals. In this study, we show that climatic scenarios of changing temperature and pH can alter the antimicrobial susceptibility profile of different *Aeromonas* species. Although species belonging to the *Aeromonas* genus are normally resistant to erythromycin and susceptible to tetracycline and sulfamethoxazole/trimethoprim, the selected strains in this study displayed variable susceptibility status to these antimicrobials; however, and with the exception of one strain (*A. hydrophila* and tetracycline), reversion of the original susceptibility status occurred for all tested strains and antimicrobial compounds at least in one experimental condition.

In some situations, reversion of non-susceptibility to susceptibility to the tested antimicrobial compounds was observed. Antibiotic resistance represents a fitness cost for bacterial species and the development of resistance is modulated by this parameter [43,44]. Resistance to antimicrobial compounds can impact important cellular activities or be met with higher energetic costs related to gene expression needs [45,46]. Thus, when experiencing amplified fitness costs, such as those provided by changes in temperature and pH, the rate of resistance reversibility in bacteria increases [43]. In this study, it seems that several combinations of water temperature and pH treatments resulted in the phenomenon that accommodates this hypothesis; however, resistance development was also observed in this study for strains displaying wild-type status. In alternative to resistance acquisition through horizontal gene transfer, a process known to be modulated by temperature conditions [42], de novo mutations (including recombination) can explain antibiotic resistance development in the absence of resistance determinants or antimicrobial pressure in the environment [45,46], as in this study. In fact, increasing temperatures have been associated with genome-wide selection of these mutations [47]. Despite the costs in fitness already described for resistance acquisition, bacterial species have the potential to downplay such costs by means of compensatory evolution by developing mutations that will decrease fitness cost without compromising antimicrobial resistance or by performing physiological adaptations or activating specific systems that buffer mutational effects and fitness costs [45,46,48,49]. Different factors can influence the acquisition of antibiotic resistance in these settings, such as thermal stress or changes in pH [50,51]. Antimicrobial resistance development occurred in this study for several combinations of water temperature and pH treatments. It is likely that the final antimicrobial susceptibility of the isolates corresponds to an “arms race” between external stressors impact, fitness costs and genetic adaptation by the bacteria, unraveling a non-linear relationship between the tested variables and the antimicrobial susceptibility of *Aeromonas* spp.

## 4. Materials and Methods

### 4.1. Strain Selection

*Aeromonas* species selection followed results obtained prior to this study [52]. Namely, the occurrence of mesophilic *Aeromonas* spp. was investigated in *Iberochondrostoma lusitanicum* in four freshwater streams in the Lisbon district, Portugal (Lizandro: 38.886701°, −9.298140°; Samarra: 38.894761°, −9.433734°; Jamor: 38.720832°, −9.249696°; Laje: 38.709159°, −9.314079°) previously characterized by our team [53]. *A. caviae*, *A. hydrophila*, *A. media* and *A. veronii* were considered the most abundant species and, hence, included in this study. Strains were selected from a bacterial library evaluated by a RAPD (random amplified polymorphic DNA) technique in order to perform molecular typing and genomic differentiation. Three isolates of each *Aeromonas* species that were not considered clones, originating from different locations, were selected as representatives for inclusion in the study (*n* = 12).

The strains’ ability to produce slime was evaluated using a phenotypical assay, Congo Red Agar (22 °C, 72 h), as described before [54]. Only slime-producer strains were selected for inclusion in the study.

Strains were stored in pure cultures in cryovials stored at −80 °C. Prior to their use, resuscitation was performed by transferring 100 µL of each bacterial suspension to 8 mL of Brain Heart Infusion broth (BHIB; VWR, Radnor, PA, USA), incubating for 24 h at 21 °C. After, bacterial suspensions were transferred to solid mediums—BHI agar and Columbia Blood (COS) agar (Biomérieux, Marcy-l’Étoile, France)—and incubated at 21 °C for 24 h. The purity of the cultures was confirmed by macro and microscopic morphology, as well as by Gram staining and phenotypic traits (oxidase production).

### 4.2. Biofilm Formation Quantification

In order to standardize the number of colony-forming units (CFU) in the suspensions to be used in the quantification of biofilm formation, reference *Aeromonas* strains were selected, namely *A. caviae* ATCC 1976, *A. hydrophila* ATCC 7966, *A. media* ATCC 33907 and *A. veronii* ATCC 35624.

Briefly, reference strains were incubated in BHI agar and COS agar at 21 °C for 24 h. After incubation, for each reference strain, colonies were selected and inoculated in 5 mL of 0.9 % saline solution until adjusting to a turbidity of 0.5 McFarland using a digital densitometer DENSIMAT (Biomérieux, Marcy-l’Étoile, France). After homogenization, serial ten-fold dilutions were performed in 9 mL of 0.9% saline solution (up to 10^−6^). From each dilution (10^−4^ to 10^−6^), 100 μL were collected and plated in BHI agar in duplicate, using sterilized glass beads. Plates were incubated at 21 °C up to 48 h. Colonies were counted in both plates and averaged. The number of CFU/mL was calculated using the formula (number of colonies × dilution factor)/volume.

Biofilm formation was performed using the microtiter plate assay and quantification was performed using the crystal violet method, as described before [55,56] with modifications. Bacterial colonies were collected from BHI agar and suspended in 5 mL of 0.9% saline solution until adjusting to a turbidity of 0.5 McFarland. Based on the pre-established average CFU/mL for each *Aeromonas* species, concentrations were adjusted for each strain in order to prepare a final concentration in the wells of the Nunc™ MicroWell™ 96-well plates (ThermoFisher Scientific^®^, Waltham, MA, USA) of 5 × 10^5^ UFC/mL in a final volume of 200 μL. As culture medium, Tryptic Soy Broth (TSB, VWR, Radnor, PA, USA) supplemented with 0.25% glucose (Millipore^®^, Merck, Darmstadt, Germany) was used. *A. hydrophila* ATCC 7966 is considered a strong biofilm producer; hence it was selected as a positive control. As a negative control, TSB supplemented with 0.25% glucose was used in six wells in each assay. The microtiter plate was incubated at 21 °C for 48 h.

After incubation, the content of all wells was carefully aspirated to eliminate planktonic forms and the wells were washed three times at room temperature with phosphate-buffered saline (PBS; VWR, Radnor, PA, USA) at pH 7.0. The PBS was discarded after the final wash and the microtiter plate was incubated in an inverted position at 60 ˚C for 1h, for the adherent cells to fixate. After, 150 μL of 0.25% Hucker crystal violet (diluted in de-ionized water; Merck, Darmstadt, Germany) were added to the wells, followed by incubation at room temperature for 5 min. The stain excess was aspirated, and the microtiter plate rinsed until the rinse was free of stain. The microtiter plate was airdried at room temperature and, once dry, 150 μL of 95% ethanol (NORMAPUR**^®^**, VWR, Radnor, PA, USA) were added to each well for solubilization of the stain. The microtiter plate was covered with the lid to avoid ethanol’s evaporation and incubated at room temperature for 30 min. After incubation, the optical density (OD) of the microtiter plate was evaluated at 570 nm in a horizontal bidirectional reading using the FLUOstar OPTIMA microplate reader (BMG LABTECH, Ortenberg, Germany). This assay was performed prior and after the microcosm assay to enable further comparisons. In both situations, three replicates were performed for each strain on independent days.

### 4.3. Antimicrobial Susceptibility Testing

Antimicrobial susceptibility testing was performed using the disk diffusion technique [57]. Guidelines of the Clinical and Laboratory Standards Institute for *Aeromonas salmonicida* testing were followed as reference [58], selected since the testing temperature—22 °C—closely resembles the temperature used for the basal treatment. The following antibiotics (Mastdiscs**^®^**, Mast Group, Liverpool, UK) were tested: erythromycin (E, 15 µg), tetracycline (T, 30 µg) and sulfamethoxazole/trimethoprim (TS, 23.75–1.25 µg). Antimicrobial compound choice followed options where epidemiological cut-off values were available. A “wild-type” (WT) phenotype implies isolate susceptibility to the antimicrobial, while a “non-wild-type” (NWT) phenotype implies that the isolate presents resistance mechanisms. *Escherichia coli* ATCC 25922 was used as a quality control. This technique was performed prior and after the microcosm assay to enable further comparisons. One strain from each species was randomly selected to be tested. Only strains from pure culture microcosms (i.e., no strains from mixed cultures microcosms were used) were used to perform the antimicrobial susceptibility testing. The same strain was used prior and after microcosm comparisons. In both situations, 10% of replicates were performed on independent days.

### 4.4. Microcosm Assay

To evaluate the influence that water temperature and pH might have in the antimicrobial resistance and virulence profiles of *Aeromonas* spp., a microcosm simulation assay was developed. Testing variables (i.e., temperature and pH) were selected based on the expected impact that climatic alterations will have in these two parameters in freshwater ecosystems [5] and on the known influence of these variables on bacterial biofilm formation and resistance acquisition/expression [42,59,60,61,62].

Regarding water temperature, four experimental conditions were used. First, a condition representing the current water temperature values was created based on trends in water temperature observed during higher temperature months (July to October) in the Lisbon’s District rivers (Cascais, Oeiras and Sintra municipalities) in the period between 1985–2016 and averaged (21 °C) [63]. Only sampling points located far from the river mouth were selected to prevent temperature oscillations related to other water bodies. Similarly, only sampling points with substantial datasets over a wide temporal frame were selected (*n* = 6). Location was selected to match the origin of the bacterial isolates. Additionally, two different 21st-century projections of climate alterations for the period of 2081–2100 establishing different levels of greenhouse gas emissions and atmospheric conditions, air pollutant emissions and land use were selected—representative concentration pathways (RCP) 4.5, representing a scenario of medium stabilization (23.2 °C) and 8.5, representing a scenario of high warming (24.5 °C) [5]. To mimic a scenario of rapid temperature fluctuations, the protocol established by Saarinen, Lindström and Ketola [22] was implemented with modifications to accommodate *Aeromonas* spp. growth conditions and the temperature ranges defined for this study. So, repetitions of 24 h cycles of either 24.5 °C or 21 °C were applied. Additionally, to establish an initial time point to enable comparisons in both the microtiter plate assay and the disk diffusion technique prior and after the microcosm assays, a treatment (T0) mimicking the current water temperature and pH (21 °C, pH 7.61) was included. Contrarily to the other treatments, the strains in T0 were incubated in river water for only 24 h.

Simulations from van Vliet et al. [7] on the correlation between air and river water temperature were used to determine final water temperature conditions for the RCP scenarios. Additionally, river discharge level, which also affects water temperature, was based on simulations by van Vliet et al. [7,14] for the Iberian Peninsula and fixed at decrease levels of 40%.

Regarding water pH, and since this parameter trends in rivers will vary according to demographic and geologic characteristics of the areas adjacent to the river [64,65], both a scenario of acidification and a scenario of alkalization were included. Three conditions were created, two mimicking both previously described scenarios and one establishing the current water pH conditions. Water pH values were established based on trends accessed in the same datasets used for temperature [63]. The treatment established as the current condition was based on the average of the values recorded in the analyzed period (pH 7.61). The acidification scenario was based on the average of the lowest pH values observed in all analyzed rivers (pH 6.31), while the alkalization scenario was based on the average of the highest pH values recorded (pH 8.61). A summary of the experimental conditions used in this study is found in Table 1.

Microcosm experimental setup was adapted from Zhang and Buckling [66] and Cairns et al. [67]. Water preparation was performed as described in Sautour et al. [33]. BHIB was used as an addictive of river’s water to act as a nutrient source. This medium was used at a 2.5% concentration to resemble the resource levels found in natural ecosystems.

Briefly, river water collected in a freshwater stream in the Lisbon district (Jamor: 38.720832°, −9.249696°) was filtered using a 0.22 µm Millipore filter (Frilabo, Maia, Portugal) and autoclaved at 121 °C for 20 min. For each water pH condition, BHIB was added to the water and pH adjusted to match the conditions established using a HI-4521 Research Grade pH/ORP/EC Bench Meter (Hanna Instruments, Póvoa de Varzim, Portugal). Bacterial suspensions were prepared by collecting colonies from BHI agar that were suspended in 5 mL of 0.9% saline solution until achieving a turbidity of 0.5 McFarland. Suspensions were prepared in pure cultures and in mixed cultures (with only one strain of each species—*A. caviae*, *A. hydrophila*, *A. media* and *A. veronii*—represented once). Nunc™ MicroWell™ 96-well plates were used to establish the microcosm. In pure culture wells, 200 μL of the respective medium were added, following the addition of 10 μL of the bacterial suspension. In the mixed culture wells, 2.5 μL of each bacterial strain was used. In both situations, bacterial suspensions were prepared in 0.9% saline solution previously according to the established average CFU/mL of the reference strains to achieve a final concentration of 5 × 10^5^ UFC/mL in each well. In the negative control wells, 210 μL of the respective medium was added. Plates were incubated for 6 days in the respective temperature treatment inside an SSI10 SSI10-2 orbital shaking incubator (Shel Lab, Cornelius, NC, USA) at 150 rpm to mimic water turbulence in the natural habitat. Every 48 h of incubation, renewal of the medium was performed by adding 20 μL of the previous culture into a new plate with 180 μL of the respective medium. At the end of each microcosm assay, the OD was read at 570 nm as described before to determine bacterial growth. After reading, 10 μL from each well was transferred into BHI agar, incubated at the respective assay’s temperature for 24 h and used for biofilm quantification, antimicrobial susceptibility testing and species confirmation (in the case of the mixed culture wells). The pH values for each assay were validated by randomly selecting bacterial cultures across the three different pH used, as well as the negative controls mediums, and analyzed using Neutralit^®^ pH-indicator paper (Merck, Darmstadt, Germany). Tests were performed immediately after incubation.

### 4.5. Aeromonas Species Confirmation in Mixed Culture Wells

Following the microcosm assays, species confirmation in the mixed culture wells was performed. Bacterial colonies with distinct macroscopic morphology in BHI agar were selected and streaked into pure cultures. The purity of the cultures was evaluated by macro and microscopic analysis, and Gram staining and oxidase production were evaluated.

Bacterial genomic DNA was obtained by the boiling method [68]. To achieve species identification, a multiplex PCR protocol previously described [69] was used with some modifications. This protocol targets the identification of the four species included in this study. *A. caviae* ATCC 1976, *A. hydrophila* ATCC 7966, *A. media* ATCC 33907 and *A. veronii* ATCC 35624 were used as positive controls.

Briefly, PCR mixtures were performed in a final volume of 25 µL, composed of: 12.15 µL of Supreme NZYTaq 2 × Green Master Mix (NZYTech, Lisbon, Portugal), 10 µL of PCR-grade water (Sigma-Aldrich, Saint Louis, MO, USA), 0.025 µL (0.05 µM) of primers A-16s, 0.25 µL (0.5 µM) of primers A-cav, 0.1 µL (0.2 µM) of primers A-med, 0.225 µL (0.45 µM) of primers A-hyd, 0.075 µL (0.15 µM) of primers A-Ver; and 1.5 µL of template DNA. Thermocycler conditions included a hot start at 95 °C for 2 min; followed by 6 cycles of denaturation at 94 °C for 40 s, annealing at 68 °C for 50 s and extension at 72 °C for 40 s; and 30 cycles at 94 °C for 40 s, 66 °C for 50 s and 72 °C for 40 s.

Amplification products were resolved by gel electrophoresis using 1.5% (*w*/*v*) agarose in 1 × TBE Buffer (NZYTech, Lisbon, Portugal). Gels were resolved for 45 min at 90 V and NZYDNA Ladder VI (NZYTech, Lisbon, Portugal) was used as a molecular weight marker. Gels were visualized using a UV light transilluminator. The images were recorded through the Bio-Rad ChemiDoc XRS imaging system (Bio-Rad Laboratories, Hercules, CA, USA).

### 4.6. Statistical Analysis

Prior to statistical analysis, the influence of the microcosm assay (i.e., other factors than the water temperature and pH conditions) on the biofilm production and antimicrobial resistance profiles was accessed by comparing the results obtained with the treatment T0 and current pH 7.61 (similar water temperature and pH conditions). A coefficient of variation of 25% was set as a breakpoint and calculated individually for each *Aeromonas* species. Minimal and maximal limits were calculated regarding T0 values. Current pH 7.61 values that fell outside the limit were considered significantly different. Replicates of isolates where this situation occurred were excluded from the subsequent analysis due to possible bias (i.e., *A. veronii* #1 3rd replicate, mixed culture #1 2nd and 3rd replicates, mixed culture #2 2nd replicate, mixed culture #3 1st and 2nd replicates). For antimicrobial resistance profiles, a qualitative comparison of the epidemiological cut-off values between the two treatments was performed and no deviations occurred.

Several isolate level response variables were analyzed regarding temperature and pH treatments. Using a factorial ANOVA where it was determined the difference in values regarding T0 treatment and Tukey’s multiple comparison test to evaluate differences between treatments, the (1) biofilm production and the (2) bacterial growth were considered. Using a stepwise linear regression and a point-biserial correlation, the influence of the different *Aeromonas* species in mixed cultures on the production of biofilm was considered. Pearson’s correlation was calculated between biofilm production and bacterial growth. The statistical analysis was performed using IBM SPSS Statistics version 27 software (IBM Analytics, New York, NY, USA). Graphs were produced using GraphPad Prism**^®^** (GraphPad Software, San Diego, CA, USA, version 5.01).

## 5. Conclusions

Current results show how *Aeromonas* spp. will respond to projected environmental shifts in water temperature and pH. Namely, that temperature increments will have a biphasic effect on *Aeromonas* spp. growth, while this bacterial genus will multiply better in acidic environments. Further, *Aeromonas* spp. biofilm production will be decreased due to temperature oscillations and microbial interactions in mixed cultures. Finally, antimicrobial resistance signatures of *Aeromonas* spp. will vary individually to changing temperature and pH parameters. Although general patterns were observed, it is evident that modulation of the intrinsic bacterial characteristics varies across isolates and that the final expression pattern will be influenced by environmental drivers and individual variability; however, the general patterns determined with this study deepen our knowledge on bacterial alterations expected in aquatic environments, strengthening our awareness and response to future bacterial outbreaks and how to deal with them.

Simplification of experimental settings, such as the approach applied in this study, has the limitation of disregarding the role of many other biotic and abiotic factors that can play a role in bacterial growth and virulence and resistance expression. Additionally, focusing on one bacterial genus to study such interactions is a major limitation of this study, since it fails to represent both the outcomes of a bacterial community that closely resembles natural communities, as well as beneficial and detrimental effects of distinct bacterial strains/species on a particular bacterial strain in focus. Further development of microcosm experiments to accommodate more complex networks of drivers and bacterial communities is required.

## Figures and Tables

**Figure 1 antibiotics-10-01008-f001:**
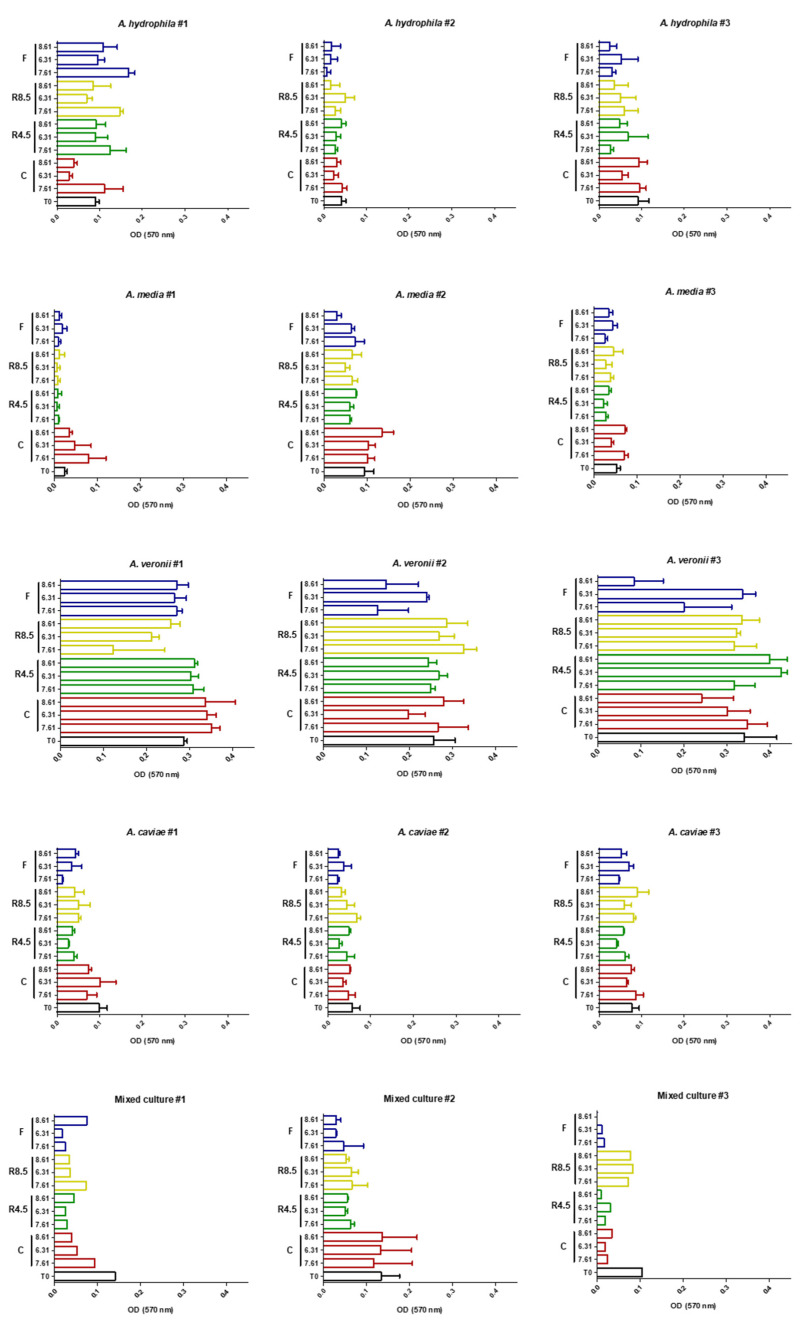
Biofilm production by each strain and mixed culture (mean + SEM). Presented results correspond to values subtracted to each treatment’s negative control for normalization. The three replicates’ results are presented by strain and mixed cultures, except for replicates where T0 and Current pH 7.61 were considered significantly different (*A. veronii* #1, Mixed cultures #1, #2 and #3). First column in each graph represents the temperature treatment (C—Current, R4.5—RCP 4.5, R8.5—RCP 8.5, F—Fluctuations) and the second the pH treatment. OD—Optical density.

**Figure 2 antibiotics-10-01008-f002:**
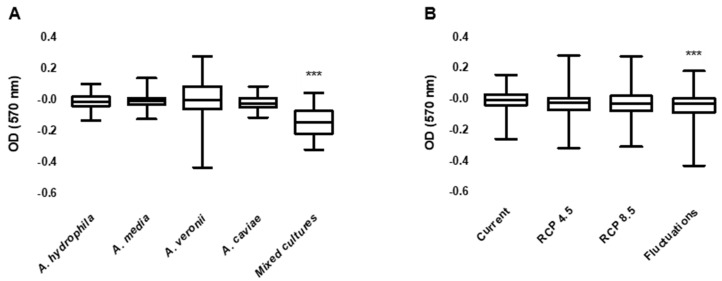
Biofilm production by individual *Aeromonas* species and mixed cultures (**A**) and in different water temperature treatment—each treatment includes all strains results. (**B**). Presented results correspond to values subtracted to each treatment’s negative control for normalization and to the corresponding T0 treatment values for comparison. OD—Optical density; *** *p* < 0.001.

**Figure 3 antibiotics-10-01008-f003:**
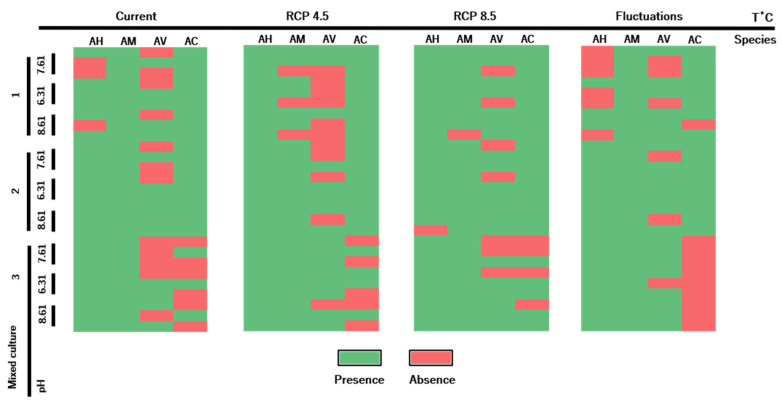
Prevalence of *Aeromonas* species in the mixed cultured wells after the microcosm assay. Each line corresponds to a distinct replicate belonging to one of the pH treatments (7.61, 6.31 and 8.61) from the tested mixed cultures (#1, #2 and #3). Each column represents an *Aeromonas* species (AH—*A. hydrophila*, AM—*A. media*, AV—*A. veronii*, AC—*A. caviae*) from a specific temperature treatment (Current, RCP 4.5, RCP 8.5 and Fluctuations).

**Figure 4 antibiotics-10-01008-f004:**
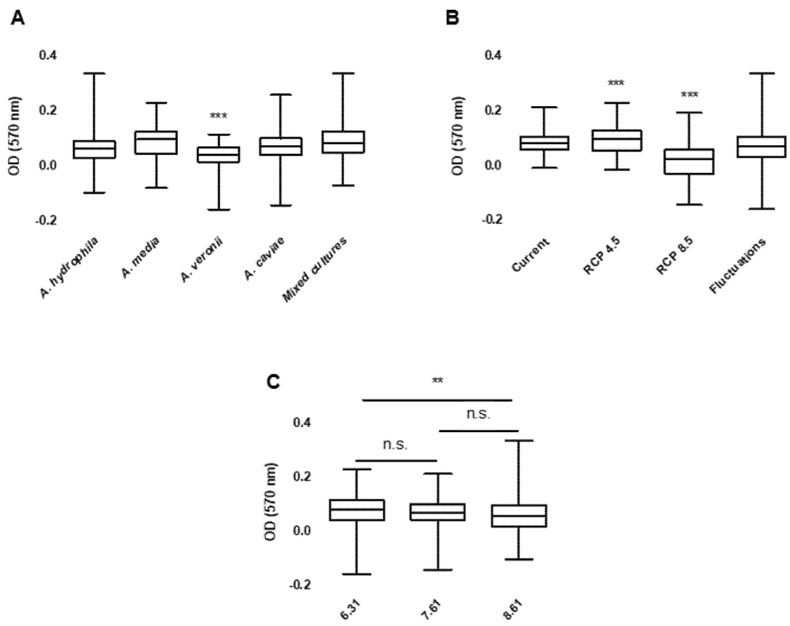
Bacterial concentration by *Aeromonas* species and mixed cultures (**A**), by water temperature treatment (**B**) and by water pH treatments (**C**). Presented results correspond to values subtracted to each treatment’s negative control for normalization and to the corresponding T0 treatment values for comparison. OD—Optical density. n.s. *p* > 0.05, ** *p* < 0.01, *** *p* < 0.001.

**Figure 5 antibiotics-10-01008-f005:**
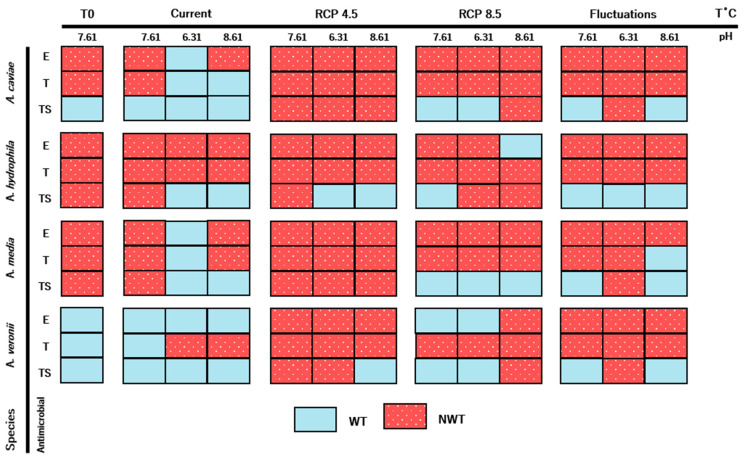
Antimicrobial resistance phenotypes (WT—wild-type, susceptible; NWT—non-wild-type, non-susceptible) of the *Aeromonas* isolates regarding water temperature and pH treatments. E—erythromycin, T—tetracycline, TS—sulfamethoxazole/trimethoprim.

**Table 1 antibiotics-10-01008-t001:** Experimental conditions used in the microcosm assays. RCP—representative concentration pathway.

Experimental Conditions
Temperature (°C)	pH
Current	21	Current	7.61
RCP 4.5	23.2	Acidification	6.31
RCP 8.5	24.5	Alkalization	8.61
Fluctuations	21–24.5

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
