# Peer review of "Climatic Alterations Influence Bacterial Growth, Biofilm Production and Antimicrobial Resistance Profiles in Aeromonas spp."

_antibiotics, 2021, doi:10.3390/antibiotics10081008_

Round 1

Reviewer 1 Report

In this manuscript, Grilo et al. have studied in Aeromonas spp changes in growth, biofilm formation and antibiotic resistance which could be induced by climatic changes. Though the authors describe large variability among Aeromonas  species and even between different isolates, they found some significant changes which can be correlated to specific microcosm conditions.

Though variability is important, significant results are clearly presented.

  1. In discussion, the authors could emphasize that the conditions they used are milder than the one used in others studied with temperature change with 10°C amplitudes or pH changes with 2 units differences.
  2. They should also discuss the limitation of their microcosm model, with only Aeromonas species either as pure culture or mixed culture. This does not reflect the complex reality with interactions between different bacteria. For instance, as shown by Messi et al (DOI: 10.1016/s0043-1354(02)00028-3), co-culture with Pseudomonas has a positive effect on survival of Aeromonas.
  3. Also, resistance to antiobiotics with temperature increase has often been shown in environment where horizontal gene transfer can occur, which is a situation different from resistance emerging from a pure culture of Aeromonas.

Author Response

We thank the reviewers for their thoughtful comments and helpful critique that have contributed to improve the manuscript. Please find below a point by point reply to the reviewers where we carefully address their concerns and inputs.

Reviewer 1:

In this manuscript, Grilo et al. have studied in Aeromonas spp changes in growth, biofilm formation and antibiotic resistance which could be induced by climatic changes. Though the authors describe large variability among Aeromonas  species and even between different isolates, they found some significant changes which can be correlated to specific microcosm conditions.

Though variability is important, significant results are clearly presented.

  1. In discussion, the authors could emphasize that the conditions they used are milder than the one used in others studied with temperature change with 10°C amplitudes or pH changes with 2 units differences.

Thank you for the interesting remark. We have now added the following sentence to the discussion (lines 154-156):

“It is noteworthy that, in this study, differences regarding growth, biofilm production and antimicrobial resistance signatures were observed using relatively small temperature and pH amplitudes, which are more likely to reflect future climatic trends.”

  1. They should also discuss the limitation of their microcosm model, with only Aeromonas species either as pure culture or mixed culture. This does not reflect the complex reality with interactions between different bacteria. For instance, as shown by Messi et al (DOI: 10.1016/s0043-1354(02)00028-3), co-culture with Pseudomonas has a positive effect on survival of Aeromonas.

In this manuscript, we are solely focusing on responses to environmental variables by the Aeromonas genus and decided not to explore the complexity of interactions with other bacterial species. We have priorly tried to address the limitation you refer in the conclusion, but we have now included further clarification so we can accommodate this concern (lines 501-509):

“Simplification of experimental settings, such as the approach applied in this study, has the limitation of disregarding the role of many other biotic and abiotic factors that can play a role in bacterial growth and virulence and resistance expression. Additionally, focusing in one bacterial genus to study such interactions is a major limitation of this study, since it fails to represent both the outcomes of a bacterial community that closely resembles natural communities, as well as beneficial and detrimental effects of distinct bacterial strains/species on a particular bacterial strain in focus. Further development of microcosm experiments to accommodate more complex networks of drivers and bacterial communities is required.”

  1. Also, resistance to antibiotics with temperature increase has often been shown in environment where horizontal gene transfer can occur, which is a situation different from resistance emerging from a pure culture of Aeromonas.

We have now revised the manuscript to include the following sentence (lines 259-264):

“In alternative to resistance acquisition through horizontal gene transfer, a process known to be modulated by temperature conditions [42], de novo mutations (including recombination) can explain antibiotic resistance development in the absence of resistance determinants or antimicrobial pressure in the environment [45,46], as in this study. In fact, increasing temperatures have been associated with genome-wide selection on these mutations [47].”

Miguel Grilo

Reviewer 2 Report

General Comments: Minor spell check/grammar needed in the Abstracts and throughout the text.

For figure 2, the authors need to mention explicitly in the legend which strains are being shown in panel B). The conclusions need to be more specific about some of the trends observed in the study.

Line 132: This is the case of erythromycin and A. caviae, A. hydrophila and A. media.

Comment: Do the authors mean erythromycin resistance; please clarify.

Line 121:   were similar to results obtained with treatment Current and similar pH levels.

Comment: with the “Current” treatment

Line 192: what can also explain differences observed not only

Comment: which can also explain

Line 205: and Aeromonas tipically increase both

Comment: Aeromonas typically increase

Line 208: before reaching maximum termal…

Comment: ……maximum thermal…

Line 219: allow for acid tolerance

Comment: The authors should include data on why this is relevant, for example, is the climate model predicting an increased acidity in general or specific cases in freshwater, along with appropriate references.

Line 257: wilt-type status

Comment: wild-type status

Line 258: de novo mutations

Comment: There is no evidence that de novo mutations increase under these conditions. The authors should reference any studies which have shown that mutations increase under such conditions. Second, did the authors perform any experiments that showed that their samples had reduced Horizontal Gene Transfer, and therefore, had to resort to recombination?

Author Response

We thank the reviewers for their thoughtful comments and helpful critique that have contributed to improve the manuscript. Please find below a point by point reply to the reviewers where we carefully address their concerns and inputs.

Reviewer 2:

General Comments: Minor spell check/grammar needed in the Abstracts and throughout the text.

For figure 2, the authors need to mention explicitly in the legend which strains are being shown in panel B).

In Figure 2, panel B, the presented graph represents the results of biofilm production by temperature treatment, pooling all the results (all strains) obtained in each treatment. In contrast, Figure 1 represents biofilm production by each strain and it is possible to compare the results with each combination of pH and temperature treatment. In order to clarify this, we have included the following sentence in Figure 2 caption (line 99):

“Biofilm production by individual Aeromonas species and mixed cultures (A) and in different water temperature’s treatment [each treatment includes all strains results] (B)”.

The conclusions need to be more specific about some of the trends observed in the study.

Thank you for your critique. We have now revised the conclusion section (lines 490-495):

“Current results show how Aeromonas spp. will respond to projected environmental shifts in water’s temperature and pH. Namely, that temperature increments will have a biphasic effect on Aeromonas spp. growth, while this bacterial genus will multiply better in acidic environments. Also, Aeromonas spp. biofilm production will be decreased due to temperature oscillations and microbial interactions in mixed cultures. Finally, antimicrobial resistance signatures of Aeromonas spp. will vary individually to changing temperature and pH parameters.”

Line 132: This is the case of erythromycin and A. caviaeA. hydrophila and A. media.

Comment: Do the authors mean erythromycin resistance; please clarify.

In this case, it refers to susceptibility. We have revised in the manuscript (line 133).

Line 121:   were similar to results obtained with treatment Current and similar pH levels.

Comment: with the “Current” treatment

Line 192: what can also explain differences observed not only

Comment: which can also explain

 Line 205: and Aeromonas tipically increase both

Comment: Aeromonas typically increase

Line 208: before reaching maximum termal…

Comment: ……maximum thermal…

Line 257: wilt-type status

Comment: wild-type status

All the changes have been made to the revised manuscript.

Line 219: allow for acid tolerance

Comment: The authors should include data on why this is relevant, for example, is the climate model predicting an increased acidity in general or specific cases in freshwater, along with appropriate references.

While most projection scenarios (and specially IPCC, 2014, which is the last report available produced by this entity) predict alterations regarding oceanic pH, with a trend of ocean acidification, the same is not possible to perform for freshwater streams on a global scale. This is due to local geochemical characteristics by the river lithology, as well as to a more vulnerable and volatile status of freshwater streams pH to anthropogenic disturbances.

This explanation was included in the Materials section, along with proper references (lines 394-396):

“Regarding water pH, and since this parameter trends in rivers will vary according to demographic and geologic characteristics of the areas adjacent to the river [64,65], a scenario of acidification and a scenario of alkalization were included.”

Line 258: de novo mutations

Comment: There is no evidence that de novo mutations increase under these conditions. The authors should reference any studies which have shown that mutations increase under such conditions. Second, did the authors perform any experiments that showed that their samples had reduced Horizontal Gene Transfer, and therefore, had to resort to recombination?

This is an interesting remark. We have now included the following sentence to reflect this concern (lines 259-264):

“In alternative to resistance acquisition through horizontal gene transfer, a process known to be modulated by temperature conditions [42], de novo mutations (including recombination) can explain antibiotic resistance development in the absence of resistance determinants or antimicrobial pressure in the environment [45,46], as in this study. In fact, increasing temperatures have been associated with genome-wide selection on these mutations [47].”

Regarding the second question, no experiments to evaluate HGT in the studied strains were performed. The hypothesis we expose in lines 259-264 are based on the assumption that HGT is not occurring due to the fact that:

  • Only microcosms in which single clones were used were selected to evaluate antimicrobial susceptibility profiles, hence new genetic material originating from other bacterial strains could not be acquired through HGT;
  • Water treatment conditions suggested by Sautour et al. (2003; https://doi.org/10.1046/j.1365-2672.2003.02048.x) and implemented in this study included both filtration and sterilization, preventing the availability and stability of microbiota and genetic material, limiting the possibility of HGT.

This was also clarified in the revised manuscript (lines 259-264).

Miguel Grilo

Reviewer 3 Report

The authors of the manuscript present work in which the idea is quite interesting however, it is not possible to talk about the influences of climatic and pH alterations on the antibiotic resistance profile not having evaluated the profile of individual strains at the end of the microcosm incubation period.
Moreover, the experimental setup and the whole methodology need to be implemented and improved.

Best regards,

Valeria Gargano

Author Response

We thank the reviewers for their thoughtful comments and helpful critique that have contributed to improve the manuscript. Please find below a point by point reply to the reviewers where we carefully address their concerns and inputs.

Reviewer 3:

The authors of the manuscript present work in which the idea is quite interesting however, it is not possible to talk about the influences of climatic and pH alterations on the antibiotic resistance profile not having evaluated the profile of individual strains at the end of the microcosm incubation period.

Thank you for your consideration. In our study, for each studied species, one strain was randomly selected to test the influence of temperature and pH microcosms on antimicrobial resistance signatures. For each species, the same strain’s profile was evaluated prior and after the microcosms’ implementation.

This methodology is explained in lines 357-359:

“This technique was performed prior and after the microcosm assay to enable further comparisons. One strain from each species was randomly selected to be tested. The same strain was used in the prior and after microcosm comparisons.”

Moreover, the experimental setup and the whole methodology need to be implemented and improved.

Although we appreciate the reviewer’s critique, we believe the experimental set-up and chosen methodology in this study are both accurate and follow techniques widely acknowledged to determine the responses pursued with this study. Moreover, we have included control critical points in the experimental set-up to ensure that the effect of confounding variables (e.g., the microcosm experiment itself, lines 466-475) could be detected and eliminated from statistical analysis. 

Miguel Grilo

Round 2

Reviewer 3 Report

The manuscript submitted by the authors is interesting, but the authors state that it was not possible at the end of the experiments:

Line 77-78 : Regarding the mixed culture wells, reisolation and identification of the initial Aeromonas pool added to each well was not possible with different combinations of temperature and pH treatments. 
In this regard for example, it is unclear how the antibiograms were performed and how they compared to those performed on the initial strains.

This part of the manuscript should be completely revised and makes it unacceptable in this version.
Perhaps the authors could consider performing an expression study on AR genes, but antibiograms cannot be performed if a pure colony cannot be obtained. 

Author Response

We thank the reviewer for the comments and helpful critique that have contributed to improve the manuscript. Please find below a point by point reply to the reviewer where we carefully address the concerns and inputs.

Miguel Grilo

Reviewer 3:

The manuscript submitted by the authors is interesting, but the authors state that it was not possible at the end of the experiments:

Line 77-78 : Regarding the mixed culture wells, reisolation and identification of the initial Aeromonas pool added to each well was not possible with different combinations of temperature and pH treatments.

In this regard for example, it is unclear how the antibiograms were performed and how they compared to those performed on the initial strains.

This part of the manuscript should be completely revised and makes it unacceptable in this version.

Perhaps the authors could consider performing an expression study on AR genes, but antibiograms cannot be performed if a pure colony cannot be obtained.

Thank you for your comments. We have revised the manuscript to further clarify the employed methodology (lines 357-362):

This technique was performed prior and after the microcosm assay to enable further comparisons. One strain from each species was randomly selected to be tested. Only strains from pure culture microcosms (i.e., no strains from mixed cultures microcosms) were used to perform the antimicrobial susceptibility testing. The same strain was used prior and after microcosm comparisons. In both situations, 10% of replicates were performed in independent days.”

The methodology employed was as follows:

  1. As a representative for each Aeromonas species in our study, one isolate per species was randomly selected to be used (line 358);
  2. Each strain was evaluated for antimicrobial susceptibility profile prior to the microcosm implementation and after the microcosm assay (line 360-361);
  3. Only strains in pure culture microcosms were used to evaluate the antimicrobial susceptibility profile (lines 359-360);
  4. After the microcosm assay, strains were inoculated in BHI agar and incubated at the respective treatment’s temperature to guarantee that the effect of the temperature would persist during colony formation to be used in AST (lines 435-438);
  5. A total of 10% of replicates was performed to guarantee the technique’s reproducibility (lines 361-362).